# Geographical epidemiology of *Hyalomma anatolicum* and *Rhipicephalus microplus* in Pakistan: A systematic review

Abrar Hussain[1]*, Sabir Hussain[2,3], Ao Yu[4], Csaba Varga[1,5], Giulio A. De Leo[4], Rebecca L. Smith[1,5]*

1 Department of Pathobiology, College of Veterinary Medicine, University of Illinois, Urbana-Champaign, IL, United States of America, 2 School of Biological, Environmental and Earth Sciences, University of Southern Mississippi, Hattiesburg, MS, United States of America, 3 Department of Infectious Diseases and Public Health, Jockey Club College of Veterinary Medicine and Life Sciences, City University of Hong Kong, Hong Kong, China, 4 Department of Earth System Science, Stanford University, Stanford, CA, United States of America, 5 Institute for Genomic Biology, University of Illinois, Urbana-Champaign, IL, United States of America

* abrarh2@illinois.edu (AH); rlsdvm@illinois.edu (RLS)

**Data Availability Statement:** All relevant data are within the paper and its Supporting Information files.

## Abstract

The livestock sector contributes almost 11% of Pakistan's GDP and is crucial to 35 million people's livelihoods. Ticks are a major economic threat, as over 80% of livestock, such as bovines, are tick-infested with *Hyalomma* and *Rhipicephalus* tick species. *Hyalomma anatolicum* and *Rhipicephalus microplus* are the most common tick species collected from livestock, transmitting primarily anaplasmosis, babesiosis, and theileriosis. We aimed to identify the geographical distribution of these two tick species and hot spot areas where the risk of these diseases being transmitted by these ticks is high. Following the PRISMA guideline, two authors conducted an independent review of literature sourced from various databases. We screened 326 research articles published between January 1, 1990, and December 31, 2023, focused on identifying the tick species at the district level. Thirty studies from 75 districts, representing 49.3% of the country's total area, detected at least one tick species through collection from animals. *R. microplus* was present in 81% (n = 61) and *H. anatolicum* in 82% (n = 62) of these sampled districts. We employed spatial and conventional statistical methods with Geographic Information Systems (GIS) after mapping the weighted distribution of both ticks (the number of ticks per standard unit of sampling effort). We identified northwestern and northcentral regions of the country as hotspots with the highest tick distribution, which aligned with the documented high prevalence of anaplasmosis, babesiosis, Crimean-Congo hemorrhagic fever (CCHF), and theileriosis in these regions. This underscores the urgent need for robust tick control measures in these districts to safeguard animal health and boost the livestock economy.

**Funding:** The author(s) received no specific funding for this work.

**Competing interests:** The authors declare that there were no conflicts of interest.

## 1. Introduction

Ticks and tick-borne diseases cost Pakistan almost US $ 13.9 to 18.7 billion in loss and damage to cattle's 3 billion pieces of hide and skin annually [1]. Ticks are a major economic threat to Pakistan's livestock sector, which contributes over 60% of the value of the agriculture sector and almost 11% of the country's GDP, with 35 million people in the rural population relying on this sector for their livelihoods [2]. This livestock produces 54 million tons of milk annually, with Buffaloes producing about 68%, while 27% is paid by cattle and 5% by sheep, goats, and camels [3].

Ticks feed themselves by sucking blood, lymph, and digested tissue from animals; this can result in the acquisition and transmission of many tick-borne pathogens (TBPs) [4]. Ticks serve as vectors and reservoirs of many pathogens of medical and veterinary concern, including several zoonotic pathogens [5]. Of the approximately 949 identified tick species worldwide, 10% are vectors of pathogens that cause diseases of public health and veterinary concern [6, 7]. Almost 30 tick species in Pakistan have been identified from cattle and buffaloes and 40 from sheep and goats [9]. In addition, more than 80% of bovines were tick-infested with *Hyalomma* and *Rhipicephalus* genera species alone. *Hyalomma anatolicum* and *Rhipicephalus microplus* are the most common tick species collected from the livestock of Pakistan and the primary vectors of the causative pathogens of babesiosis, theileriosis, and anaplasmosis in ruminants [1]. *R. microplus* (cattle tick) is a vector for *Anaplasma marginale*, *Babesia bigemina*, and *B. bovis*, causative agents of tick fever in livestock. *H. anatolicum* is a proficient vector for *Theileria annulata*, which causes theileriosis, resembling malaria in animals [8]. *Hy. anatolicum* is also responsible for transmitting Crimean-Congo Hemorrhagic Fever Virus (CCHFV), which results in a disease (CCHF) with a 30% fatality rate [9]. Zoonotic tick-borne diseases (TBDs), especially anaplasmosis, babesiosis, and ehrlichiosis, can be transmitted to humans via tick bites during practices like removing ticks from animal bodies or crushing of removed ticks with bare hands and possibly through contact with blood or tissues of infected animals [10].

Various factors influence the spatial distribution of ticks. They thrive within specific humidity (60–80%) and temperature (27–39˚C) ranges. Pakistan is situated in a subtropical zone, providing favorable climate for the growth of ticks and the transmission of TBPs [2]. Physiography, soil type, and land use are also important determinants of ticks' presence and distribution. *Rhipicephalus microplus* is globally distributed in subtropical and tropical regions [11], and the distribution of *H. anatolicum* is high in relatively hot and dry climates, such as semi-desert steppes, savannas, and Mediterranean scrubland [12]. The number of potential host species ticks may feed upon to complete their life cycle and might also influence their distribution. Ticks that feed upon multiple animal hosts with different mobility, such as birds, might have a wider spatial distribution than ticks that feed only on one host. *R. microplus* is a one-host tick. Eggs hatch into larvae in a few weeks, and these larvae seek a host (i.e., cattle), attach themselves, and blood-feed. During their attachment and feeding, larvae molt into nymphs, and nymphs develop into adults. Adults drop off the host, and the female lays eggs to start the cycle again [13]. In contrast, *H. anatolicum* can be a two-host or three-host tick, with larval and nymphal blood meals from small mammals and birds and adult blood from cattle, humans, and other large mammals (Fig 1) [14, 15].

Despite the importance of *H. anatolicum* and *R. microplus* in Pakistan, limited information is available on their geographical distribution. Therefore, this study performed a systematic review of literature to describe the geographical distribution of these two tick species and to identify potential transmission hot spots where tick abundance per standard unit of sampling effort is higher than expected. The provided data will aid animal health stakeholders in better understanding the spatial variability of TBDs risk among Pakistan's livestock to design effective prevention and control strategies.

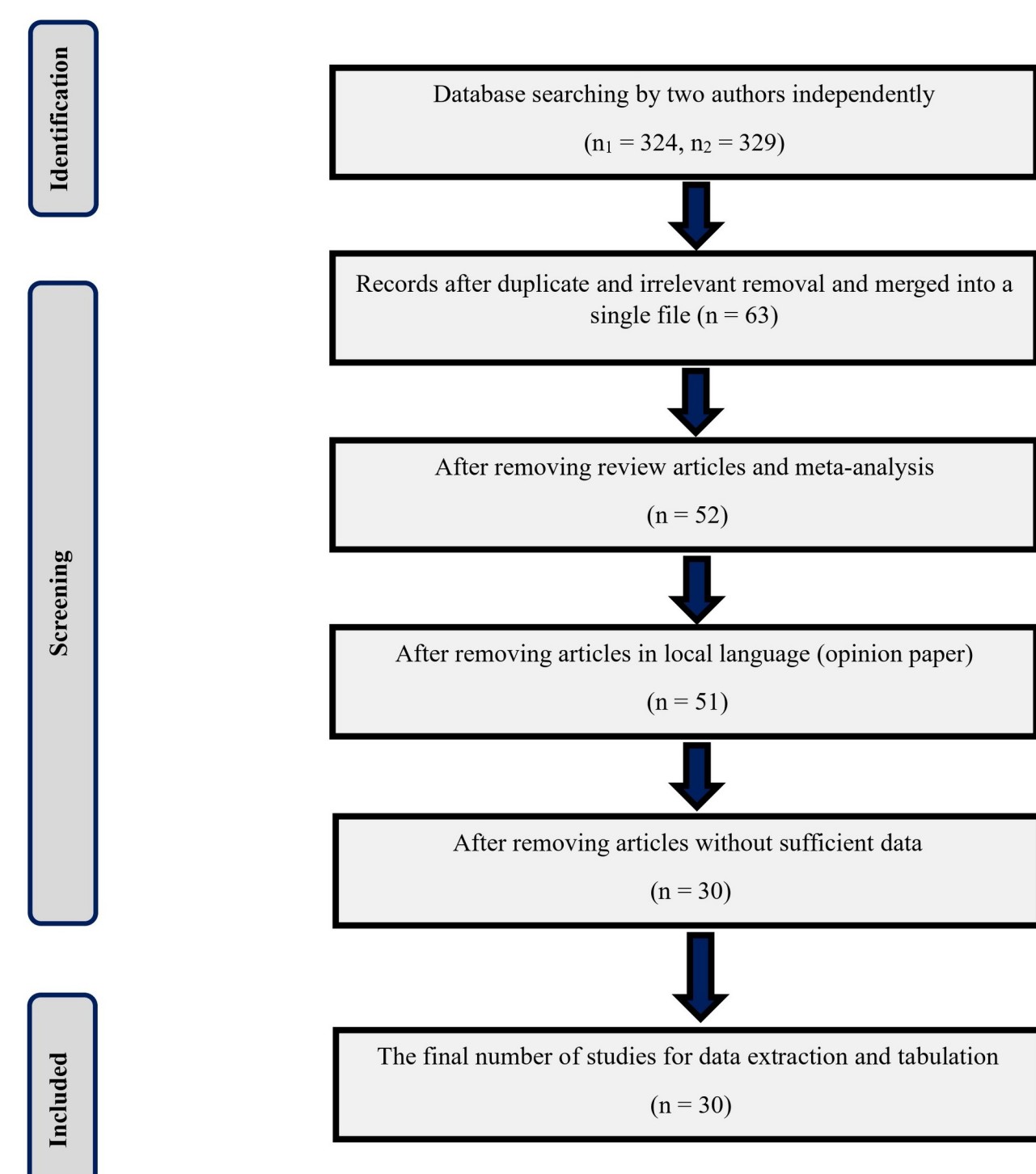

**Fig 1. Schematic overview of the literature search procedure and results (PRISMA 2020 checklist is available as supporting information in S1 Checklist).**

## 2. Materials and methods

### 2.1. Review protocol

We conducted a systematic review following the Preferred Reporting Items for Systematic Reviews and Meta-Analyses (PRISMA) guidelines [16] on the distribution of *H. anatolicum* and *R. microplus* in Pakistan.

### 2.2. Data sources and search strategy

During the data retrieval phase, we employed three search engines, Scopus, PubMed, and Web of Science, between October 1, 2023, and January 31, 2024, to identify articles related to *H. anatolicum* and *R. microplus* in Pakistan. The search encompassed articles published from January 1, 1990, to December 31, 2023. The keyword combination used for the search for the final search string was: *Hyalomma anatolicum* OR *Rhipicephalus microplus* OR cattle tick OR tick AND Pakistan. The University of Illinois Urbana-Champaign (UIUC) library utilized the online search database (S1 File).

### 2.3. Data extraction

Two authors (A.H. and S.H.) independently extracted and compiled search information in separate spreadsheets. Each author screened the data to eliminate duplicate studies. Subsequently, the datasets were merged to prevent redundancy. Any disparities in the extracted data between the two authors were cross-verified and discussed to generate comprehensive lists of relevant articles. These lists included authors, study title, year of publication, journal name, volume, issue, page number, DOI, author affiliations, abstract, and keywords.

### 2.4. Study selection criteria

During the screening process, titles, abstracts, and keywords were utilized to eliminate duplicates, literature review studies, and studies not published in English (S2 File). Subsequently, a comprehensive review of the full text of all studies was conducted to both screen papers for inclusion criteria and to extract essential details. The inclusion criteria were: (i) inclusion of the identification of *H. anatolicum* and/or *R. microplus*, (ii) provision of details regarding the time of sample collection, (iii) specification of the location of sample collection sites (to at least at the district level, the third-level administrative division of Pakistan), and (iv) mention of the techniques employed for identification.

### 2.5. Quality assessment and selection

Data from studies passing the screening process were organized in tabular form using Microsoft Excel (365). During data extraction, we recorded only the general tick numbers, disregarding their developmental stages, to ensure consistency across different districts. This approach was necessary because only a few papers provided detailed information on tick life stages. The details extracted encompassed the duration of sample collection, study title, study location, tick identification methods, the total number of ticks collected, the individual counts of *H. anatolicum* and *R. microplus* ticks collected, pathogens detected from both ticks (if any), and the reference of the study (S3 File).

### 2.6 Risk of bias assessment

Search, screening and data extraction were performed independently by two authors working in different academic institutions and validated by a third author before data analysis and visualization.

The Corresponding author of this paper crosschecked the inclusion and exclusion criteria and consistency of retrieved information to reduce the risk of bias and also did the risk of bias assessment using an appraisal tool for cross-sectional studies (AXIS) included studies (S4 File).

## 2.7 Data visualization

We used R Studio version 4.1.3 for data visualization and ArcGIS 10.7.1 (Environmental Systems Research Institute, Inc., Redlands, CA, USA) to map the geographical distribution of tick species. We made the following simplifying assumptions to map infestation and effort data:

- The study's sampling effort was averaged over the study's time.

- For studies spanning more than one district, sampling effort was assumed to be even across sampled districts if not specified.

- If districts were aggregated within a study, we assumed that infestation was homogenous across sampled districts.

- If year but not month was specified, we spread sampling effort and infestation levels homogeneously across the year.

## 2.8 Spatial analysis and mapping

The Incremental Spatial Autocorrelation (Global Moran's I) Tool was utilized to assess the global clustering of tick infestation and both tick species by examining a series of incrementally increasing distances. This process involved evaluating the strength of global spatial clustering at each distance increment [17]. The peak distance with the highest global clustering was used for the local cluster analysis.

Hot spot analysis of district-level tick distribution for each tick species was conducted using the Getis-Ord Gi* method [18]. Each district was represented by its centroid and the weighted number of tick species for this analysis. The null hypothesis assumes complete spatial randomness of districts with high or low reported numbers of both tick species. The statistics provide a Z-score and p-value associated with the standard normal distribution. High (positive) or low (negative) Z-scores, associated with small p-values, are observed in the tails of the normal distribution and indicate a statistically significant pattern. An important ($p \leq 0.05$) hotspot area is indicated by a high positive Z-score ($\geq +1.96$), where districts with a high number of ticks are surrounded by districts also with a high number of ticks. Similarly, a statistically significant cold spot ($p \leq 0.05$) area is indicated by a high negative Z-score ($\leq -1.96$) and is detected if districts with a low number of ticks are surrounded by districts also with a low number of ticks. To account for zoning differences (e.g., two districts might have the same number of cases, but the district shapes and boundaries differ) and edge effects (e.g., districts at the periphery of the study area have neighboring areas that are not included in the study region), the "zone of indifference" conceptualization parameter was employed.

## 3. Results

### 3.1 Literature search results

The PRISMA flow diagram indicates the selection process for eligible studies (Fig 2). Two authors independently retrieved 328 and 339 studies, respectively, from different databases. After removing duplicates and articles not featuring the specified tick species, 63 articles remained. Subsequently, we excluded 11 articles lacking original research content (e.g., review articles and meta-analyses), and in language-based screening, one non-English study (an opinion article) was eliminated. Additionally, we removed 21 studies that lacked sufficient data,

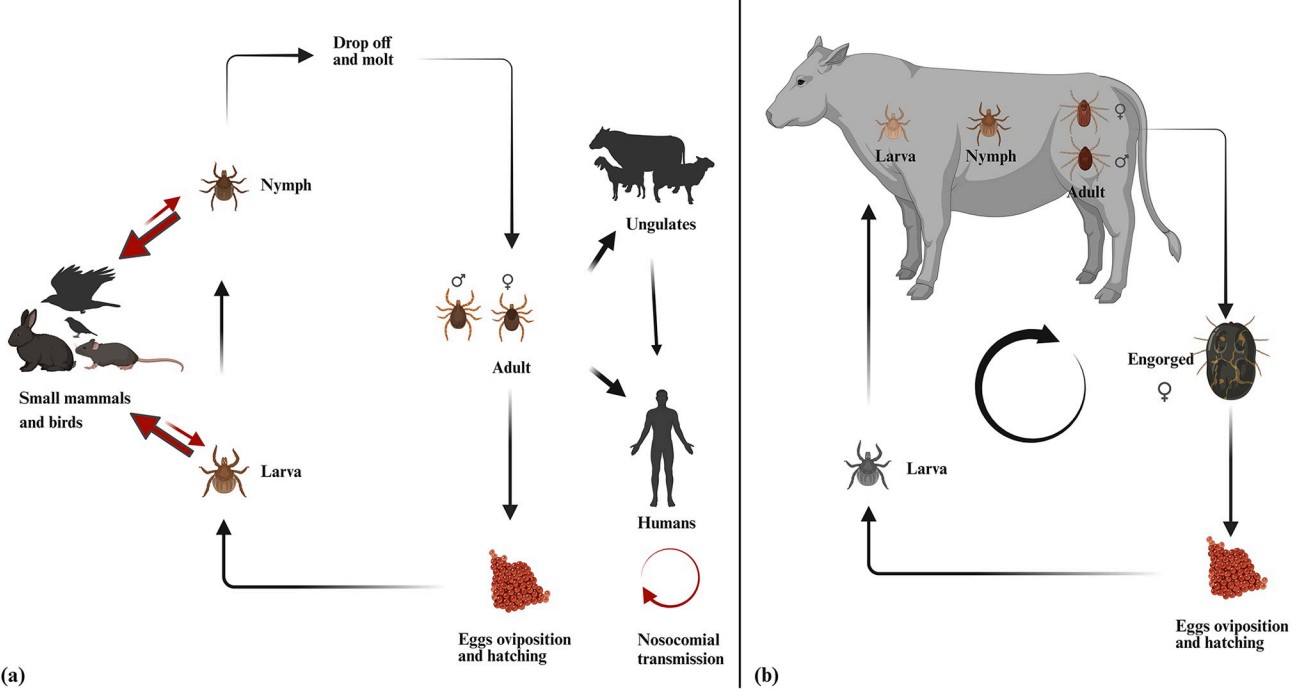

**Fig 2.** (a) Life cycle of *Hyalomma anatolicum* (b) Life cycle of *Rhipicephalus microplus* (Figure Made in BioRender.com).

such as geographical information, sampling information, and conference proceedings, with insufficient details. The final selection comprised 30 studies.

### 3.2 Geographic coverage of ticks

Among the 30 included studies, only four relied on morphological identification methods for ticks, while the other 26 used both morphological and molecular identification methods (PCR). The most common study design (n = 21 studies, 70%) was convenience sampling for tick collection from animals, followed by two-stage cluster-randomized sampling (n = 9, 30%). A total of 75 districts (44.12% of the 170 districts in Pakistan, encompassing 49.3% of the 880,940 km$^2$ country area) were sampled for both ticks: 30 from the province of Punjab, 27 from KPK, 12 from Baluchistan, three from Gilgit Baltistan, and one from Islamabad (Fig 3A). At least one tick species was identified in 72 of the 75 districts; the exceptions were Killa Abdullah, Ziarat, and Khyber Agency. Sampling was performed on a variety of mammals (cattle, buffalo, sheep, goat, camel, donkey, horse, mule, and dog), but more than 80% of ticks were collected from ruminants (cattle, buffaloes, sheep, and goats). Animal infestation rate was reported in 59 of the 75 districts and ranged from 14% to 86%. The highest tick infestation rates were reported in Quetta, Upper Dir, Buner, and Bajaur agencies (Fig 3B). Among the 75 studied districts, 36 were sampled repeatedly across multiple studies and years, including Attock, Chakwal, Charsadda, Peshawar, and Rawalpindi, where tick infestation data were gathered up to five times; for those, we calculated the average infestation across studies.

### 3.3 Distribution of ticks

To account for differences in the duration of sampling effort when analyzing the spatial distributions of ticks over the 75 districts, we derived "the number of ticks per standard unit of sampling effort" as the ratio between the sum of the average number of *H. anatolicum* and *R.*

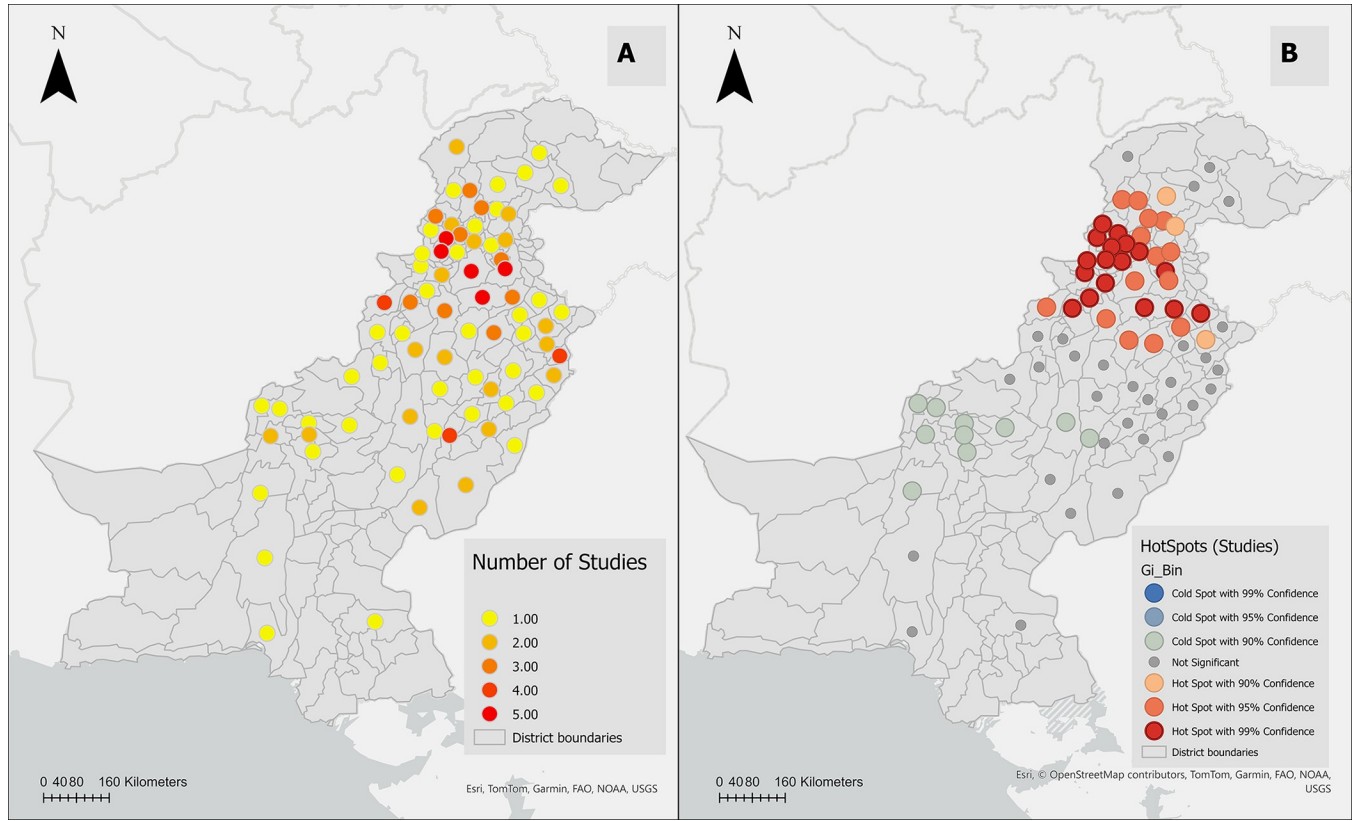

**Fig 3.** **(A)** A cartographic representation of study locations for *Hyalomma anatolicum* and *Rhipicephalus microplus* in Pakistan. **(B)** A cartographic representation of the prevalence of animals infested with *Hyalomma anatolicum* or *Rhipicephalus microplus* in Pakistan.

*microplus* ticks reported in each study in each district and the total number of months spent on sampling in the district (Fig 4).

**3.3.1 *H. anatolicum*.** From the 75 districts, 20,042 ticks were collected from different animals for identification, of which 10,996 were *H. anatolicum* from 62 (82%) of the districts. The majority of *H. anatolicum* (5,196) were collected from Khyber Pakhtunkhwa (KPK), followed by Punjab (4977), Baluchistan (711), Gilgit Baltistan (GB, 112), and Sindh (96) (Fig 5). The weighted average of *H. anatolicum* varied among districts, with Gilgit (n = 512), Mansehra n = 334), Khanewal (n = 334), Batagram (n = 320), Okara (n = 309), Diamir (n = 264), Abbottabad (n = 234), Kohistan (n = 200), Dera Ismail Khan (n = 158), and Vehari (n = 138) being at the top (Fig 7A).

To determine the extent of clustering of the high number of *H. anatolicum* in Pakistan, the Incremental Spatial Autocorrelation (Global Moran's I) Tool was used. One peak (corresponding to the maximum Z-score) was identified at 273.5 km (Fig 6).

The hot spot analysis identified three districts (Bannu, North Waziristan, Mianwali) with a low number of *H. anatolicum* (cold spots) in the west-central region of Pakistan and seven districts (Chitral, Gilgit, Diamir, Astor, Upper Dir, Swat, Kohistan) with a high number of *H. anatolicum* (hot spots) in north-western parts of the country (Fig 7B).

Only a small number of studies, 30% (9/30) from 37/75 districts (49.3%), reported tick infection prevalence and identified the pathogen species. Pathogen testing of *H. anatolicum* found CCHFV in 20 districts, *Theileria annulata was* reported in two districts, and *A. ovis*, *Ehrlichia spp.*, *Rickettsia slovaca*, and *R. massiliae* were reported in one district (Table 1).

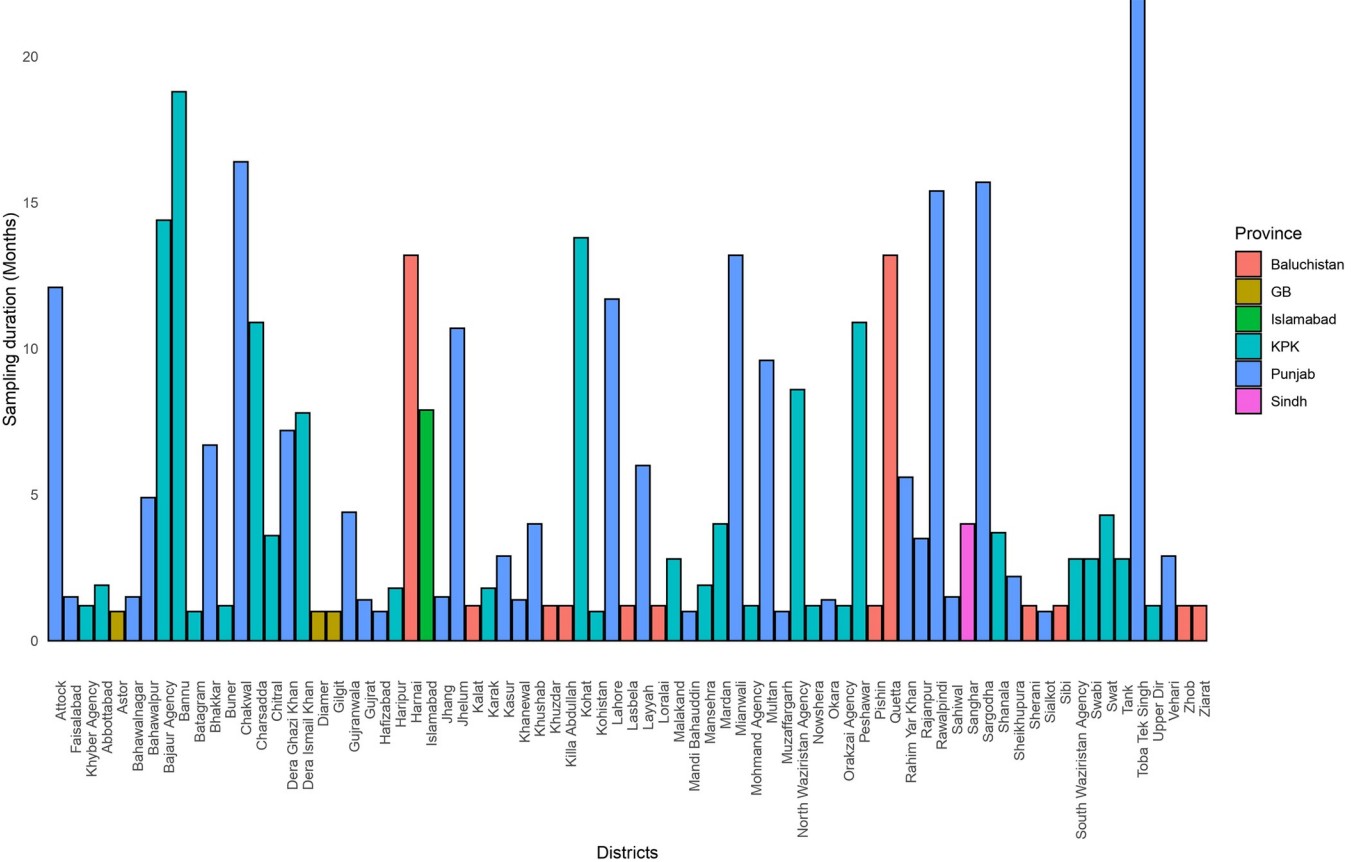

**Fig 4. The average sampling duration of *Hyalomma anatolicum* and *Rhipicephalus microplus* in the different districts of Pakistan (KPK = Khyber Pakhtunkhwa, GB = Gilgit Baltistan).**

**3.3.2 *R. microplus*.** A total of 9,046 *R. microplus* was detected in 81% (n = 61/75) of the total districts with animal sampling. By province, the majority of *R. microplus* were from Punjab (4244, 47%) or Khyber Pakhtunkhwa (KPK, 4158, 46%), followed by Baluchistan (353, 4%), Islamabad (140, 2%), Gilgit Baltistan (GB, 95, 1%), and Sindh (36, 0.4%) (Fig 5). When weighted for sampling effort, average numbers of *R. microplus* varied by district, led by Gilgit (n = 432), Batagram (n = 296), Mansehra (n = 244), Diamir (n = 216), Kohistan (n = 192), Abbottabad (n = 164), Astor (n = 112), and Gujranwala (n = 107) (Fig 9A).

To determine the extent of clustering of the high number of *R. microplus* in Pakistan, the Incremental Spatial Autocorrelation (Global Moran's I) Tool was used. One peak (corresponding to the maximum Z-score) was identified at 269.8 km (Fig 8).

The hot spot analysis identified ten districts (Loralai, Zhob, Sherani, Dera Ghazi Khan, Dera Ismail Khan, Muzaffargarh, Layyah, Khushab, Khanewal, Multan) with a low number of *R. microplus* (cold spots) in south-western parts of Pakistan and 14 districts (Chitral, Buner, Shangla, Batagaram, Haripur, Abbottabad, Mansehra, Gilgit, Diamir, Astor, Upper Dir, Swat, Kohistan, Islamabad) with a high number of *R. microplus* (hot spots) in the north-central region of the country (Fig 9B).

Detection of CCHFV and *A.marginale* was reported from *Rhipicephalus microplus* in 10 districts, *A. ovis*, *A. centrale*, *R. aeschlimannii*, *R. massiliae*, and *R. slovaca* were in 4 districts, *Theileria annulata* was reported in two districts, while *Babesia occultans*, *B. bovis* and *B. bigemina* were detected only in one district (Table 2).

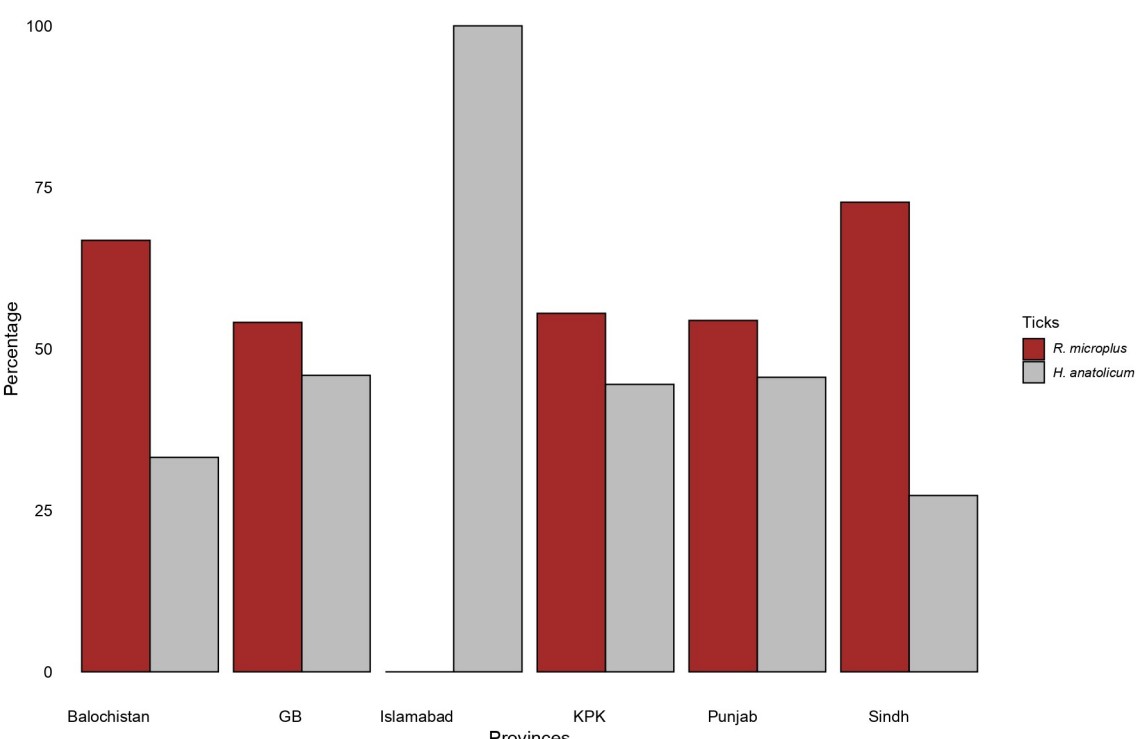

**Fig 5. The prevalence of *Hyalomma anatolicum* and *Rhipicephalus microplus* infestation reported on animals in different provinces of Pakistan (KPK = Khyber Pakhtunkhwa, GB = Gilgit Baltistan).**

## 4. Discussion

This study performed a systematic review of the literature and compiled and spatially analyzed the available data on the distribution, relative abundance, and geographical clustering of Pakistan's two most common vector tick species, *Hyalomma anatolicum* and *Rhipicephalus microplus*. In addition, the reported presence of tick-borne pathogens (TBPs) in these two tick species was summarized. Two provinces, Punjab and KPK, were highly represented in the surveillance literature, while the other two provinces, Baluchistan and Sindh, lacked data on a majority of their districts. The lack of research in these regions does not indicate the absence of these tick species or diseases transmitted by them; instead, it may be caused by a lack of resources in those regions for tick and tick-borne disease research. These two provinces have fewer universities and research institutes, especially veterinary universities, as only three universities are located in Sindh and only one in Baluchistan compared to 10 in Punjab and 5 in KPK.

To characterize the distribution of both tick species, we employed a stepwise analysis combining spatial and conventional statistical methods with Geographic Information Systems (GIS). Initially, we constructed tick-distribution maps to visually represent the geographic distribution of the prevalence of *H. anatolicum* and *R. microplus* using reported data from nationwide sources. Given that multiple studies have been conducted in some districts, we normalized the number of ticks by sampling duration (as a proxy for sampling effort) to ensure comparability. Our analysis revealed an uneven distribution of both tick species across various districts. In the northern and central regions of the country, the burden of *H. anatolicum* and *R. microplus* was notably higher compared to other regions. Both ticks were highly prevalent in Gilgit, Mansehra, Diamir, Batagram, and Abbottabad, while *H. anatolicum* was highly

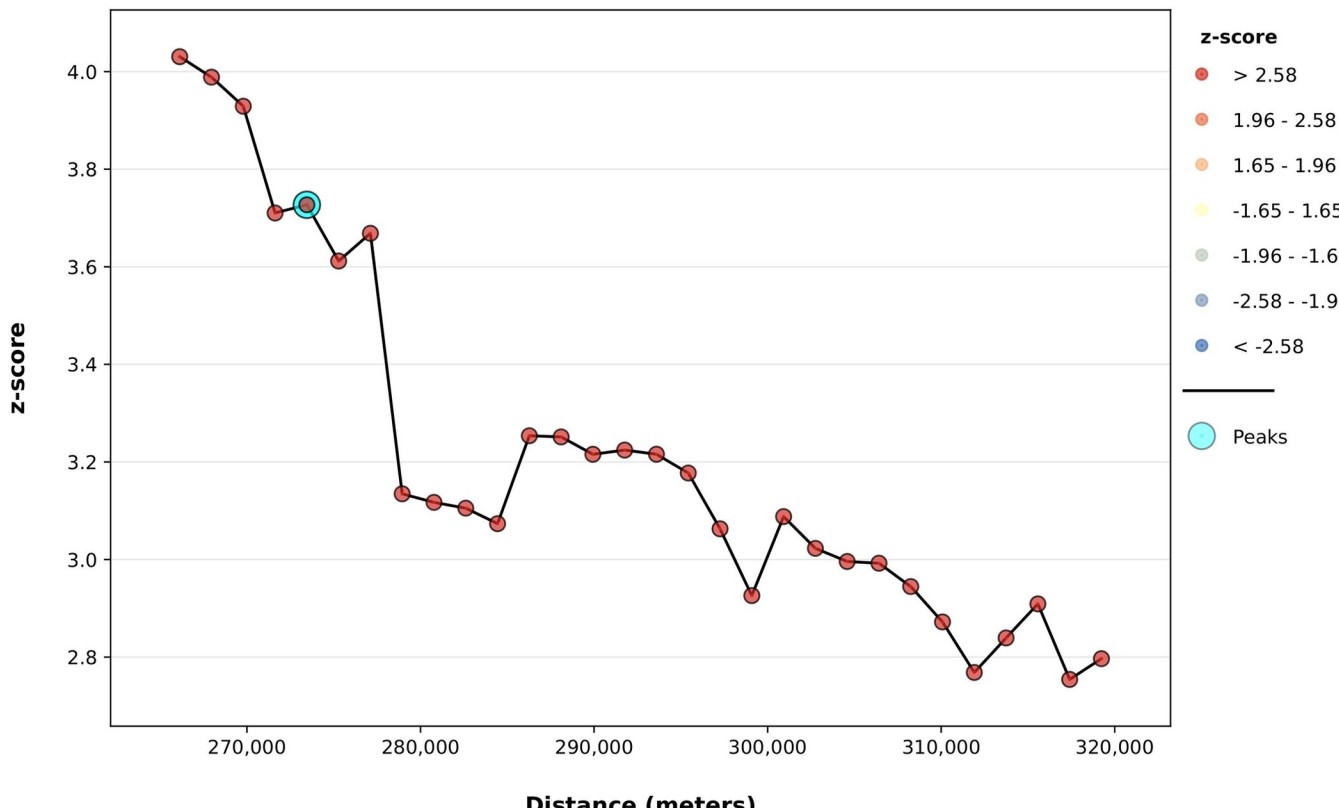

**Fig 6. Incremental spatial autocorrelation analysis for the *H. anatolicum*.** (The Global Moran's I statistic results were obtained using a default incremental distance set to the average distance to each district's nearest neighboring centroid. The color assigned to each point on the graph represents the statistical significance of the z-score values. The peak signal indicates the distance at which the spatial processes influencing clustering are most prominent. The zone of indifference conceptualization parameter was applied in the analysis, with statistical significance set at p ≤ 0.05).

prevalent in Khanewal, and Okara and *R. microplus*, was highly prevalent in Kohistan, Astor, and Gujranwala. These districts are in distinct climatic regions and belong to different provinces (Punjab and KPK), where livestock practices and tick-control measures vary.

The Incremental Spatial Autocorrelation (Global Moran's I) Tool was employed to assess individually the global clustering of two tick species distributions across thirty increasing distance bands. This evaluation of global spatial autocorrelation took into account the locations of districts as well as their respective tick distributions. The analysis utilized the concept of the "zone of indifference," which posits that all districts within a specific distance band receive maximum weighting, with a rapid decline in weighting beyond this distance as the influence level decreases [29, 30]. The highest values of Moran's I Index, z-score, and p-value were observed at distances of 273.5km for *H. anatolicum* and 269.8km for *R. microplus*, indicating that both tick species were widely distributed across numerous areas throughout the country.

During the second stage of our spatial analysis, we utilized the Hot Spot Analysis (Getis-Ord Gi*) method. This approach identifies areas with high tick distribution (hot spots, where districts with similarly high rates surround a district with high tick distribution) and detects areas with low tick distribution (cold spots, where districts with similarly low rates surround a district with low tick distribution) [31]. As shown in the hot/cold spot map, the northwestern part of the country, including Chitral, Gilgit, Diamir, Astor, Upper Dir, Swat, and Kohistan, was identified as statistically significant hot spots for *H. anatolicum*. Similarly, the north-central part of the country, including Chitral, Buner, Shangla, Batagaram, Haripur, Abbottabad,

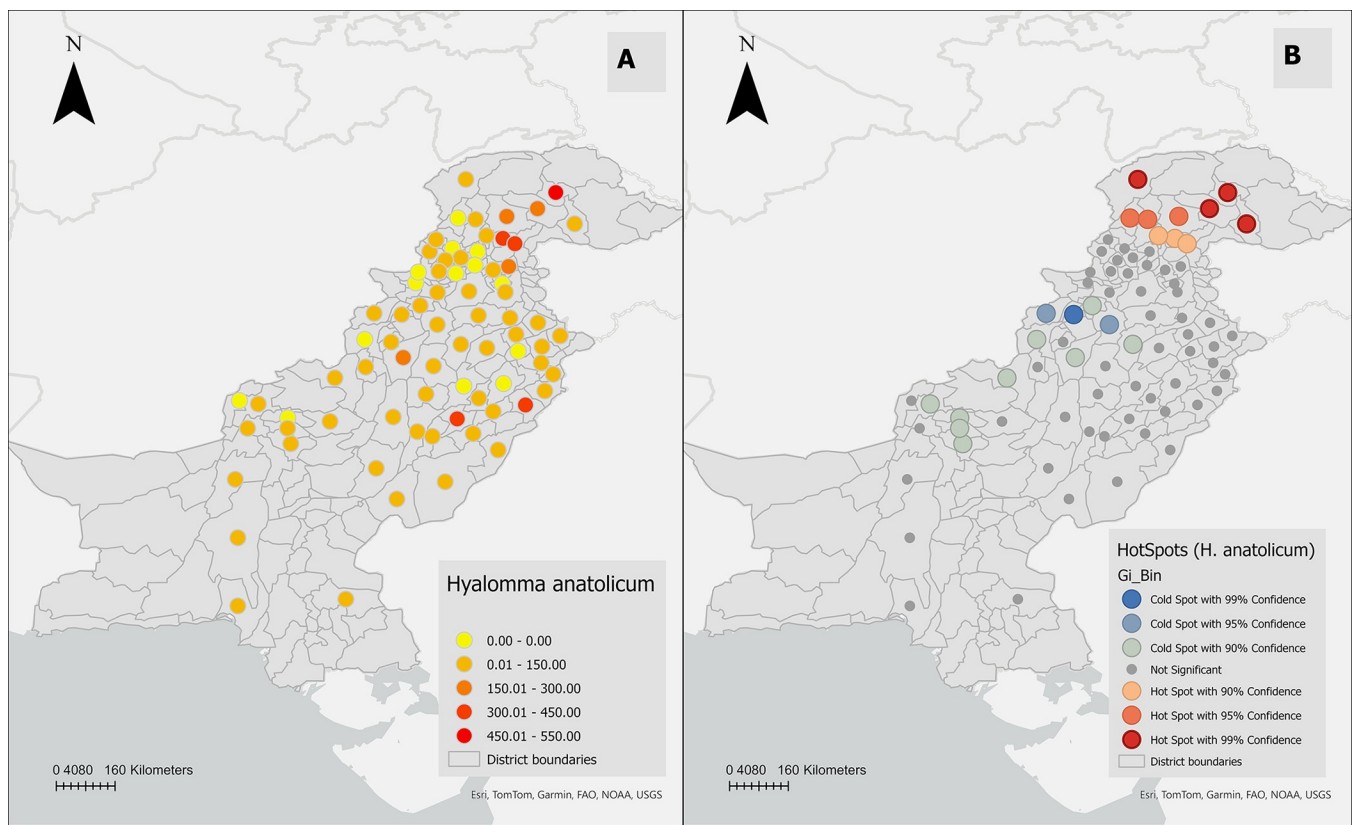

**Fig 7.** **(A)** The weighted average distribution of *Hyalomma anatolicum* among the districts of Pakistan (the number of ticks collected was divided by the number of sampling months) **(B)** Hot-spot analysis of *Hyalomma anatolicum* among the districts of Pakistan by using Getis-Ord Gi* statistic (Hot spots (depicted in red) districts with high infestation prevalence surrounded by neighboring districts also exhibiting high infestation prevalence. Conversely, cold spots (shown in blue) indicate districts with low infestation prevalence surrounded by neighboring districts with similarly low infestation prevalence. A Euclidean distance band of 273.5 km and a conceptual parameter for the zone of indifference were employed for the analysis, with statistical significance set at $p \leq 0.05$).

Mansehra, Gilgit, Diamir, Astor, Upper Dir, Swat, Kohistan, and Islamabad, was recognized as hot spots for *R. microplus*, consistent with our previous tick distribution mapping results. In contrast, the west-central part of the country, including Bannu, North Waziristan, and Mianwali, was identified as cold spots for *H. anatolicum*. Meanwhile, the southwestern part of the country, including Loralai, Zhob, Sherani, Dera Ghazi Khan, Dera Ismail Khan, Muzaffargarh, Layyah, Khushab, Khanewal, and Multan, was identified as cold spots for *R. microplus*. The

**Table 1. Pathogens detected from *Hyalomma anatolicum* collected from different districts.**

| Pathogens | Districts | References |
|---|---|---|
| Crimean–Congo hemorrhagic fever virus (CCHFV) | Attock, Rahim Yar Khan, Chakwal, Rajanpur, Rawalpindi, Mianwali, Lahore, Bahawalpur, Harnai, Kalat, Khuzdar, Killa Abdullah, Lasbela, Loralai, Pishin, Quetta, Sherani, Sibi, Zhob, and Ziarat | [19–22] |
| *Theileria annulata* | Layyah and Bajaur Agency | [23, 24] |
| *Anapalsma ovis* | Mohmand Agency | [25] |
| *Ehrlichia spp.* | Mohmand Agency | [25] |
| *Rickettsia slovaca* | Mohmand Agency | [25] |
| *R. massiliae* | Mohmand Agency | [25] |

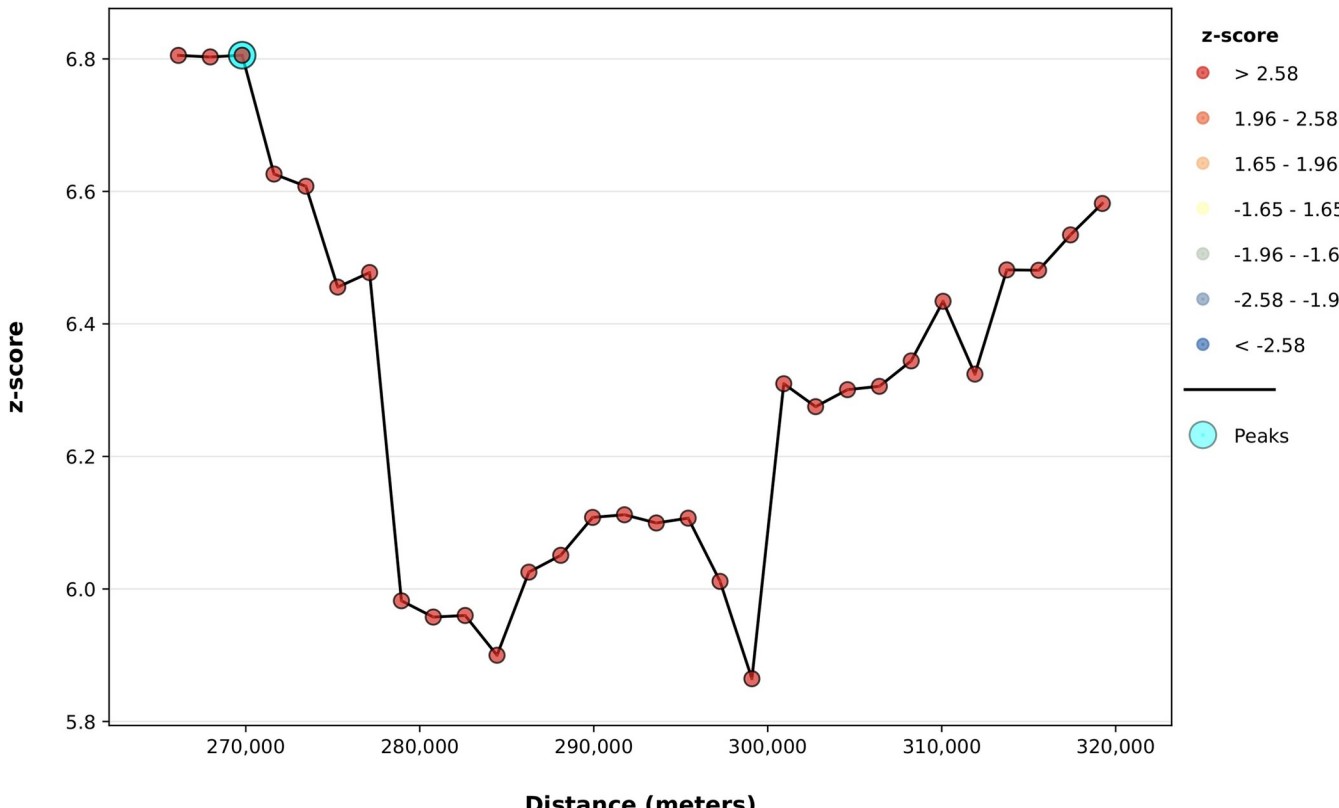

**Fig 8. Incremental spatial autocorrelation analysis for the *R. microplus*.** (The Global Moran's I statistic results were obtained using a default incremental distance, which was set to the average distance to each district's nearest neighboring centroid. The color assigned to each point on the graph represents the statistical significance of the z-score values. The peak signal indicates the distance at which the spatial processes influencing clustering are most prominent. The zone of indifference conceptualization parameter was applied in the analysis, with statistical significance set at p ≤ 0.05).

hot spot analysis may be subjected to some bias because there is still a significant region of the country where tick surveillance studies have not been conducted. However, based on the available data, this study identified regions as hotspots where future studies should be conducted to assess the factors impacting the higher-than-expected distribution of ticks. It's important to note that cold spots represent areas where the relative distribution of ticks is low, but this doesn't imply the absence of these two tick species in these regions when compared with tick-distribution maps.

Many pathogens, including parasites, viruses, and bacteria, have been reported globally as being transmitted or potentially transmitted by *H. anatolicum* and *R. microplus* with animal reservoirs, and some can be zoonotic [32]. *H. anatolicum* is the main vector of *T. annulata*, *T. lestoquardi*, *T. ovis*, *T. equi*, and *B. caballi*, as well as the CCHFV [15, 33], while *R. microplus* is a vector of *A. phagocytophilum*, *A. marginale*, *B. bigemina*, *B. bovis*, *E. chaffeensis*, *E. canis*, various species of *Rickettsia*, and *Borrelia*. In addition to transmitting pathogens and causing substantial blood loss, these ticks also inflict severe wounds during their bites [34, 35]. In our study, although we found both ticks have contributed toward all major disease-causing pathogens, *H. anatolicum* was primarily reported positive for theileriosis and CCHF-causing pathogens, while *R. microplus* was more commonly found to be positive for babesiosis and anaplasmosis-causing pathogens.

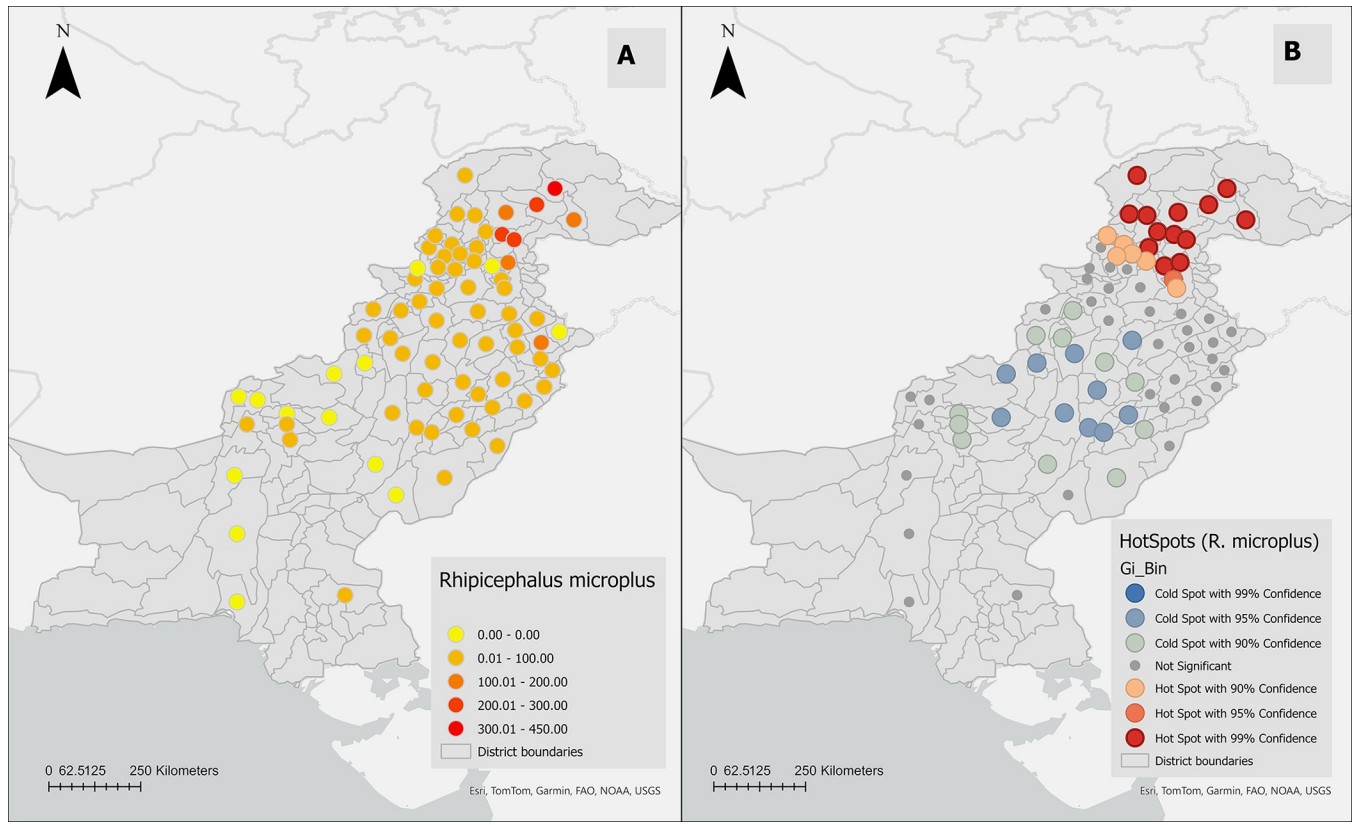

**Fig 9.** **(A)** A cartographic representation of the weighted average distribution of *Rhipicephalus microplus* among the districts of Pakistan. **(B)** Hot-spot analysis of *Hyalomma anatolicum* among the districts of Pakistan by using Getis-Ord Gi* statistic(Hot spots (depicted in red) indicate districts with high infestation prevalence surrounded by neighboring districts also exhibiting high infestation prevalence. Conversely, cold spots (shown in blue) indicate districts with low infestation prevalence surrounded by neighboring districts with similar infestation prevalence. A Euclidean distance band of 269.8 km and a conceptual parameter for the zone of indifference were employed for the analysis, with statistical significance set at $p \leq 0.05$).

**Table 2. Pathogens detected from *Rhipicephalus microplus* collected from different districts.**

| Pathogens | Districts | References |
|---|---|---|
| Crimean–Congo hemorrhagic fever virus (CCHFV) | Attock, Rahim Yar Khan, Rajanpur, Rawalpindi, Chakwal, Mianwali, Lahore, Jehlam, Dera Ghazi Khan, and Bahawalpur | [22] |
| *Theileria annulata* | Layyah and Bajaur Agency | [23, 24] |
| *Anaplasma marginale* | Charsadda, Chitral, Mardan, North Waziristan Agency, Peshawar, Shangla, South Waziristan Agency, Swat, and Tank | [26, 27] |
| *Babesia occultans* | Bajaur Agency | [23] |
| *Babesia bovis* | Kohat | [28] |
| *Babesia bigemina* | Kohat | [28] |
| *Anaplasma ovis* | Bajaur Agency, Mohmand Agency, Orakzai Agency, and North Waziristan Agency | [25] |
| *Anaplasma centrale* | Bajaur Agency, Mohmand Agency, Orakzai Agency, and North Waziristan Agency | [25] |
| *Rickettsia aeschlimannii* | Bajaur Agency, Mohmand Agency, Orakzai Agency, and North Waziristan Agency | [25] |
| *R. massiliae* | Bajaur Agency, Mohmand Agency, Orakzai Agency, and North Waziristan Agency | [25] |
| *R. slovaca* | Bajaur Agency, Mohmand Agency, Orakzai Agency, and North Waziristan Agency | [25] |

### 4.1 Theileriosis

We found that the theleriosis-causing pathogens from both tick species, H. anatolicum and R. microplus, were in Punjab and KPK. Theileriosis is an economically impactful disease in cattle, sheep, and goats and infects wild ruminants [8]. It can be transmitted by several ticks, including *Amblyomma*, *Haemaphysalis*, *Hyalomma*, and *Rhipicephalus*, but in Africa and Asia, it is mainly transmitted by ticks of the genus *Hyalomma* [36]. This aligns with our finding that *H. anatolicum* frequently carries the pathogens of this disease in Pakistan. Recent reports have found *T. annulata* in animal blood samples from the districts of Chitral and Upper Dir [37], Astor [38], and Swat [39], all of which lie within the hotspot regions identified by our study, which supports the role of these ticks in transmitting theileriosis in these regions.

### 4.2 Crimean–Congo hemorrhagic fever virus (CCHFV)

We found that CCHFV was reported only in *H. anatolicum* from various districts of Baluchistan, while in some districts of Punjab, CCHFV was reported in both *R. microplus* and *H. anatolicum*. Overall, *H. anatolicum* has been the more commonly reported carrier of this deadly virus in Pakistan [22]. Although CCHF cases have been reported across the country, CCHF cases cluster in the country's southwestern region, in the province of Baluchistan [40–42], supporting the role of *H. anatolicum* in transmitting CCHFV in this region. *Hyalomma* ticks play a critical role in the maintenance of CCHFV-endemic foci in nature, and it has been suggested that an increase in the population of *Hyalomma* ticks is followed by a rise in CCHFV infections in humans in the affected area [43]. CCHF cases among humans have also previously been reported in the districts of Kalat and Quetta, where we found CCHFV-positive ticks reported [41, 44]. This region is near the border with Afghanistan, which is also endemic for CCHF [44, 45]. Baluchistan lies in the trade path of importation of livestock from Afghanistan and ruminants' skins and hides from Iran, which is also endemic for CCHFV [46]. Chitral, Buner, Shangla, Batagaram, Haripur, Abbottabad, Mansehra, Gilgit, Diamir, Astor, Upper Dir, Swat, Kohistan, and Islamabad.

### 4.3 Babesiosis

Babesiosis is a zoonotic disease of global concern, with a growing focus on this disease within human medicine [47]. In other parts of the world, this disease is transmitted by *R. microplus* [48] along with *I. ricinus* [49]. We found two species of *Babesia* (*B. bovis* and *B. bigemina*) reported in *R. microplus* in Pakistan; both species can cause Babesiosis in ruminants. These two pathogens have been detected in the blood samples of the ruminant population in different districts of Pakistan, especially Buner [50] and Islamabad [51]' which were found to be hotspots in this study. Babesiosis is considered an endemic disease in ruminant populations of many regions of Pakistan, with reports of drug resistance [52] raising concerns about control of this pathogen. In this scenario, tick-control measures could play an important role in minimizing disease incidence in the livestock population.

### 4.4 Anaplasmosis

Anaplasmosis is common in tropical and subtropical regions of the world, particularly bovine anaplasmosis is caused mainly by *A. marginale* and, to a lesser extent, by *A. centrale* [53]. Globally, these pathogens are commonly transmitted by *Hyalomma* and *Rhipicephalus* spp. [54]. We find that both pathogens have been reported in *R. microplus* in Pakistan, while *A. ovis* (which is responsible for causing babesiosis in the ovine) has been reported in *H. anatolicum*. Bovine infection with *Anaplasma marginale* has been reported in several districts, including

Islamabad [51] and many other districts in hotspots of *H. anatolicum*. Anaplasmosis can be challenging to differentiate clinically from theileriosis or babesiosis [55], a situation worsened by the lack of clinical experts and diagnostic facilities in the less developed districts of Pakistan. Animals recovering from this disease frequently develop an asymptomatic carrier status [54], which can maintain the diseases in herds and devastate the livestock economy.

This study has systematically reviewed all the literature in the previous 33 years, providing district-level abundance and testing of *H. anatolicum* and *R. microplus*, Pakistan's two most common vector tick species. Our analysis of *H. anatolicum* and *R. microplus* distribution in Pakistan identified northwestern and northcentral regions with increased tick distribution of both ticks, respectively. This information helps the livestock department and stakeholders to develop and enhance the tick-control measures for reducing TBDs in livestock and public health risks. There is still a significant region of the country where tick surveillance studies have not been conducted, which should be prioritized for future efforts to determine the true extent of these vector ticks and their pathogens.

## Supporting information

**S1 Checklist. PRISMA 2020 checklist.**
(DOCX)

**S1 File. Studies identified in the literature search.**
(XLSX)

**S2 File. Excluded studies.**
(XLSX)

**S3 File. Included studies, with data extraction.**
(XLSX)

**S4 File. Risk of bias assement of included studies.**
(XLSX)

## Author Contributions

**Conceptualization:** Abrar Hussain, Rebecca L. Smith.

**Data curation:** Abrar Hussain, Sabir Hussain.

**Formal analysis:** Abrar Hussain.

**Methodology:** Abrar Hussain, Csaba Varga.

**Supervision:** Rebecca L. Smith.

**Visualization:** Abrar Hussain, Ao Yu.

**Writing – original draft:** Abrar Hussain.

**Writing – review & editing:** Sabir Hussain, Ao Yu, Csaba Varga, Giulio A. De Leo, Rebecca L. Smith.

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
