## [Decision Letter · Decision Letter 0]

15 Jul 2024

PONE-D-24-24395Geographical epidemiology of Hyalomma anatolicum and Rhipicephalus microplus in Pakistan: a systematic reviewPLOS ONE

Dear Dr. Hussain,

Thank you for submitting your manuscript to PLOS ONE. After careful consideration, we feel that it has merit but does not fully meet PLOS ONE’s publication criteria as it currently stands. Therefore, we invite you to submit a revised version of the manuscript that addresses the points raised during the review process.

**The manuscript needs editing revision beside the authors need to clarify the following points:**

**1. Geographical nature of each tick species distribution (like; desert, forest etc.....) **

**2. Photos for male and females of both ticks**

**3. Stages of ticks on the ground due to H. anatolicum is multiple host while R. microplus is one host**

We look forward to receiving your revised manuscript.

Kind regards,

Shawky M Aboelhadid, PhD

Academic Editor

PLOS ONE

Journal Requirements:

Reviewers' comments:

Reviewer's Responses to Questions

**Comments to the Author**

1. Is the manuscript technically sound, and do the data support the conclusions?

Reviewer #1: Yes

2. Has the statistical analysis been performed appropriately and rigorously? 

Reviewer #1: Yes

3. Have the authors made all data underlying the findings in their manuscript fully available?

Reviewer #1: Yes

4. Is the manuscript presented in an intelligible fashion and written in standard English?

Reviewer #1: Yes

5. Review Comments to the Author

**Reviewer #1:** Kindly check the grammatically error and the references carefully. The current manuscript is suitable for publication. This is meta analysis of prevalence CCHF and both species of ticks in Pakistan regions.

6. PLOS authors have the option to publish the peer review history of their article (what does this mean?). If published, this will include your full peer review and any attached files.

Reviewer #1: **Yes: **Dr Sachin Kumar

---

## [Author Response · Author response to Decision Letter 0]

29 Jul 2024

Response to the Reviewer

Editor

1. The geographical nature of each tick species distribution (like; desert, forest etc.....) 

Thank you!

A brief geographical distribution of both tick species has been added (Line #72-75)

2. Photos for male and females of both ticks

Thank you!

In the life cycle (Figure 2), images of both male and female ticks have been added.

3. Stages of ticks on the ground due to H. anatolicum being multiple host while R. microplus is one host

Thank you for your suggestion.

During data extraction, we recorded only the general tick numbers, disregarding their developmental stages, to ensure consistency across different districts. This approach was necessary because only a few papers provided detailed information on tick stages. (Line #128-130)

Reviewer

Reviewer #1: Kindly check the grammatically error and the references carefully. The current manuscript is suitable for publication. This is meta analysis of prevalence CCHF and both species of ticks in Pakistan regions.

Thank you for endorsing the manuscript for publishing. The manuscript has been revised by a native English speaker, and references have been formatted according to journal requirements.

---

## [Editor Report · Decision Letter 1]

31 Jul 2024

Geographical epidemiology of Hyalomma anatolicum and Rhipicephalus microplus in Pakistan: a systematic review

PONE-D-24-24395R1

Dear Dr. Abrar Hussain,

We’re pleased to inform you that your manuscript has been judged scientifically suitable for publication and will be formally accepted for publication once it meets all outstanding technical requirements.

Kind regards,

Shawky M Aboelhadid, PhD

Academic Editor

PLOS ONE
---

## [Editor Report · Acceptance letter]

14 Aug 2024

PONE-D-24-24395R1 

PLOS ONE

Dear Dr. Hussain, 

I'm pleased to inform you that your manuscript has been deemed suitable for publication in PLOS ONE. Congratulations! Your manuscript is now being handed over to our production team.

Kind regards, 

on behalf of

Professor Shawky M Aboelhadid 

Academic Editor

PLOS ONE